# Current Applications of Raman Spectroscopy in Intraoperative Neurosurgery

**DOI:** 10.3390/biomedicines12102363

**Published:** 2024-10-16

**Authors:** Daniel Rivera, Tirone Young, Akhil Rao, Jack Y. Zhang, Cole Brown, Lily Huo, Tyree Williams, Benjamin Rodriguez, Alexander J. Schupper

**Affiliations:** 1Department of Neurosurgery, Icahn School of Medicine at Mount Sinai, One G. Levy Place, New York, NY 10029, USA; Daniel.rivera@icahn.mssm.edu (D.R.); akhil.rao@icahn.mssm.edu (A.R.); jack.zhang@icahn.mssm.edu (J.Y.Z.); cole.brown@icahn.mssm.edu (C.B.); lily.huo@icahn.mssm.edu (L.H.); benjamin.rodriguez@icahn.mssm.edu (B.R.); alexander.schupper@mountsinai.org (A.J.S.); 2Sinai BioDesign, Department of Neurosurgery, Mount Sinai, New York, NY 10029, USA; williams.d.tyree@gmail.com; 3Rensselaer Polytechnic Institute, Troy, NY 12180, USA

**Keywords:** Raman spectroscopy, neurosurgery, intraoperative guidance, tumor margin differentiation, real-time tissue characterization

## Abstract

Background: Neurosurgery demands exceptional precision due to the brain’s complex and delicate structures, necessitating precise targeting of pathological targets. Achieving optimal outcomes depends on the surgeon’s ability to accurately differentiate between healthy and pathological tissues during operations. Raman spectroscopy (RS) has emerged as a promising innovation, offering real-time, in vivo non-invasive biochemical tissue characterization. This literature review evaluates the current research on RS applications in intraoperative neurosurgery, emphasizing its potential to enhance surgical precision and patient outcomes. Methods: Following PRISMA guidelines, a comprehensive systematic review was conducted using PubMed to extract relevant peer-reviewed articles. The inclusion criteria focused on original research discussing real-time RS applications with human tissue samples in or near the operating room, excluding retrospective studies, reviews, non-human research, and other non-relevant publications. Results: Our findings demonstrate that RS significantly improves tumor margin delineation, with handheld devices achieving high sensitivity and specificity. Stimulated Raman Histology (SRH) provides rapid, high-resolution tissue images comparable to traditional histopathology but with reduced time to diagnosis. Additionally, RS shows promise in identifying tumor types and grades, aiding precise surgical decision-making. RS techniques have been particularly beneficial in enhancing the accuracy of glioma surgeries, where distinguishing between tumor and healthy tissue is critical. By providing real-time molecular data, RS aids neurosurgeons in maximizing the extent of resection (EOR) while minimizing damage to normal brain tissue, potentially improving patient outcomes and reducing recurrence rates. Conclusions: This review underscores the transformative potential of RS in neurosurgery, advocating for continued innovation and research to fully realize its benefits. Despite its substantial potential, further research is needed to validate RS’s clinical utility and cost-effectiveness.

## 1. Introduction

Neurosurgery stands at the forefront of medical precision, where the stakes are exceptionally high due to the complex and delicate nature of the brain and its eloquent structures. In this field, the margin for error is minimal, and achieving optimal outcomes depends not just on meticulous surgical technique but also on the surgeon’s ability to accurately distinguish between healthy and pathological tissues during an operation. The challenges inherent in neurosurgical procedures drive the continuous search for innovations that enhance both visual and molecular distinctions, thus enabling more effective and safer surgical interventions.

One of the promising innovations in this context is Raman spectroscopy (RS). Although the Raman effect was first discovered by C.V. Raman in 1928, earning him the Nobel Prize in Physics in 1930, its application in neurosurgery is a recent development. For decades, RS was primarily used in chemistry and materials science. It was not until the late 20th and early 21st centuries that advances in laser technology, detectors, and data processing made it feasible for biomedical applications. The first applications of RS in neurosurgery emerged in the early 2000s, with pioneering studies demonstrating its potential for distinguishing normal brain tissue from tumors [1].

The evolution of RS in neurosurgery has been marked by significant milestones. In 2015, Jermyn et al. reported the first use of RS in living human brain tissue, achieving 90% accuracy in discriminating normal brain tissue from cancer [1]. This breakthrough paved the way for further clinical applications. By 2017, portable clinical Stimulated Raman Scattering (SRS) systems for intraoperative ex vivo neuropathology were introduced, capable of creating interpretable SRS histology mosaics in about 2.5 min [1].

RS’s ability to differentiate between tissue types based on their biochemical signatures is crucial for neurosurgeons (Figure 1) [1]. It assists in accurately identifying tumor margins and critical structures, thus enhancing surgical precision [2]. The use of RS in an intraoperative setting is particularly valuable as it provides immediate and precise biochemical characterization of tissues without the need to send samples to a pathologist for analysis. This real-time information enables surgeons to make informed decisions during critical neurosurgical procedures, potentially reducing operative time and improving surgical outcomes [3].

Recent studies have demonstrated the high sensitivity and specificity of RS in glioma detection. For instance, a study using the extreme Gradient Boosted trees (XGB) and Support Vector Machine with Radial Basis Function kernel (RBF-SVM) methods achieved 87% accuracy in distinguishing IDH-mutant from IDH-wildtype gliomas using RS data [4]. Another study, using Principal Component Analysis–Linear Discriminant Analysis (PCA-LDA) on fresh tissue samples, achieved even higher accuracy with 91% sensitivity and 95% specificity in predicting glioma IDH subtypes [4].

By eliminating the delay associated with the traditional histopathological examination, RS empowers neurosurgeons to adjust their surgical strategy on the fly, optimizing the extent of resection while minimizing damage to healthy tissues. The rationale for integrating RS into neurosurgical operations extends beyond its non-invasiveness and ability to provide real-time data under sterile conditions. This technique has shown remarkable potential in detecting malignancies, guiding surgical resections, and verifying the completeness of tumor removal [5,6,7].

The integration of machine learning algorithms with RS data has significantly improved the accuracy of tissue classification and tumor margin detection. For example, a CNN trained on 2.5 million SRS images was able to distinguish not only tumor tissue with high accuracy but even the main histopathological classification of brain tumors [1]. This demonstrates the potential for RS to provide rapid, automated diagnoses during surgery.

However, it is important to note that while RS shows great promise, its clinical translation still faces challenges. These include the need for standardization of data acquisition and analysis methods, as well as the requirement for larger clinical trials to validate its efficacy in various neurosurgical applications [4]. Additionally, the complexity of biological Raman data necessitates sophisticated data processing techniques, which are still evolving [1].

This literature review will evaluate the current research landscape concerning RS as a surgical adjunct in neurosurgery. It will focus on how this technique contributes to improved surgical outcomes and assess its comparative efficacy against traditional methods. Through a detailed analysis, this review will explore the operational integration, technological advancements, and practical challenges of RS in the neurosurgical arena, providing insights into its future applications and development.

## 2. Materials and Methods

A comprehensive literature review was conducted by two independent reviewers to gather and analyze data concerning real-time RS. The methodology for this systematic review adhered to the Preferred Reporting Items for Systematic Reviews and Meta-Analyses (PRISMA CRD42024581777) guidelines [Figure 2] [8].

We utilized PubMed as the database for our data extraction, searching for peer-reviewed journal articles published from the inception of the database through 13 March 2024. Employing a MEDLINE search strategy guide, we employed Medical Subject Headings (MeSH) terms to refine our search, specifically designed to capture a broad spectrum of studies related to RS in intraoperative neurosurgery [9]. Specific search terms used included combinations of terms related to brain cancer, brain tumors, intracranial neoplasms, neuro-oncology, neurosurgical oncology, epilepsy, functional neurosurgery, pediatric neurosurgery, gliomas, and meningiomas. Additionally, terms related to intraoperative techniques and Raman spectroscopy, such as “Raman histology”, “Raman scattering”, “Coherent anti-Stokes Raman scattering”, “Stimulated Raman Histology”, and “Raman imaging”, were included. Boolean operators (AND, OR) were employed to refine the search outcomes.

The inclusion criteria focused on original research articles that discussed real-time applications of RS with fresh human tissue samples in or near the operating room. This criterion was vital to ascertain the direct applicability and impact of the findings in a clinical context. Excluded from this review were retrospective studies, review articles, case reports, letters to the editor, conference abstracts, non-human studies, correspondences, and any articles that lacked full-text availability.

Titles and abstracts were screened to remove irrelevant papers, duplicates, and those not meeting the inclusion criteria. The remaining articles were subjected to a full-text review to further assess their suitability based on specific relevance to the neurosurgery context, thus providing a thorough overview of the current landscape.

QUADAS-2 was chosen as the quality assessment tool for this study due to its specific design for diagnostic accuracy studies. QUADAS-2 is tailored to evaluate studies that compare an index test to a reference standard for diagnosing a particular condition. This makes QUADAS-2 particularly well-suited for assessing studies like the one by Jabarkheel et al. [10], which examines the diagnostic accuracy of Raman spectroscopy in identifying pediatric brain tumors. QUADAS-2 allows for a comprehensive evaluation of potential biases in patient selection, index test conduct and interpretation, reference standard application, and patient flow and timing—all critical aspects in diagnostic accuracy studies. Furthermore, QUADAS-2 includes an assessment of applicability, which helps determine how well the study’s results might generalize to the review question at hand. This feature is especially valuable when synthesizing evidence from multiple diagnostic accuracy studies, as it helps reviewers understand not just the internal validity of each study but also how relevant its findings are to the specific clinical context being investigated.

The total number of citations for all studies was obtained from public-domain citation data from the National Institute (NIH) Open Citation Collection [11]. The relative citation ratio (RCR) of each eligible manuscript was queried from the NIH of Health iCite website to evaluate the influence of each article relative to other NIH-funded studies [11,12].

## 3. Results

From an initial pool of 861 articles screened, 37 studies met the inclusion criteria (Figure 2). These selected studies provide a comprehensive overview of how Raman spectroscopy and stimulated Raman histology are being increasingly integrated into neurosurgical practice for real-time tumor diagnostics. Conducted between 2015 and 2023, the studies primarily focus on intraoperative tumor margin differentiation, with the added benefit of machine learning and artificial intelligence models. The median RCR among these studies is 2.26, with a median citation count of 14. Collectively, the studies encompass over 2000 patients, with a median sample size of 26, representing a blend of small exploratory cohorts and extensive clinical trials. The USA, Canada, and Germany are the leading contributors, with the USA producing most of the high-impact research. The study designs are diverse, ranging from prospective observational trials and case–control studies to large-scale clinical trials, addressing key applications such as meningioma grading, glioma margin detection, and enhanced surgical decision-making via real-time imaging. The global research effort highlights the expanding clinical interest in these technologies, driven by their potential to improve surgical outcomes through rapid, label-free diagnostics and precise tumor resection, especially in complex cases like diffuse gliomas and skull base tumors.

### 3.1. Quality Assessment

The QUADAS-2 risk of bias assessment for the studies included in this review revealed variability across the four key domains: patient selection, index test, reference standard, and flow and timing (Figure 3). In terms of patient selection, most studies presented unclear risks, as they did not provide detailed information on whether patients were consecutively enrolled or if exclusion criteria were clearly defined. For the index test domain, unclear risk was consistently observed due to a lack of blinding in the interpretation of the index test results, particularly in studies that utilized Raman spectroscopy or multimodal fiber–probe spectroscopy. The reference standard domain exhibited a high risk of bias in several studies, as histopathological evaluations were typically conducted without blinding to the index test results, raising concerns about interpretation bias. In contrast, the flow and timing domain generally presented low risk across most studies, as appropriate time intervals between the index tests and reference standards were maintained, and no significant patient exclusions or missing data were reported. Overall, the findings indicate that while the flow and timing were well-controlled, patient selection and blinding in both the index test and reference standard remain critical areas for improvement.

### 3.2. Raman Spectroscopy Mechanism of Action

RS utilizes monochromatic light from near-infrared (NIR), visible, or ultraviolet (UV) ranges to analyze materials through the Raman effect, an inelastic light scattering phenomenon [13]. When monochromatic light interacts with a sample, most photons scatter elastically without energy change, known as Rayleigh scattering [37,38]. However, a small fraction of photons undergo inelastic scattering, experiencing a shift in energy due to interactions with the vibrational modes of molecules within the sample [39].

This energy modification can result in photons gaining energy (anti-Stokes shift) or losing energy (Stokes shift) [40]. These shifts are unique to the specific molecular vibrations and chemical bonds present in the material, providing a distinctive molecular fingerprint in the form of a Raman spectrum [31,40]. The spectrum reveals peaks corresponding to the vibrational frequencies of the molecular bonds, offering insights into the sample’s molecular structure and composition [41]. This process, though rare compared to Rayleigh scattering, yields highly specific information about the biochemical makeup of the sample, making Raman spectroscopy a powerful tool for material analysis [42].

### 3.3. Instrumentation and Technology

RS encompasses a variety of optical techniques that harness the light–matter interaction to provide detailed insights into the chemical and structural composition of brain tissue. These methods are each underpinned by distinct physical principles and technological implementations. By manipulating how light interacts with molecular vibrations, RS techniques can elucidate the molecular foundations of complex systems, from minute chemical compounds to complex intracranial pathologies, offering invaluable tools for diagnosis, research, and quality control.

#### 3.3.1. Surface-Enhanced Raman Spectroscopy

Surface-enhanced Raman spectroscopy (SERS) is a sensitive technique that amplifies Raman signals using nanostructured surfaces, often of metals like gold or silver [43,44]. The primary enhancement mechanism, electromagnetic enhancement, involves the metal nanoparticles which, when illuminated, create localized surface plasmons that intensify the electromagnetic field and thereby boost the Raman signal of nearby molecules [45,46]. A secondary mechanism, chemical enhancement, occurs when molecules adsorbed on the metal surface participate in charge transfers that alter their electronic states and enhance Raman scattering [47]. SERS substrates are engineered to maximize these effects by optimizing nanoparticle size, shape, and arrangement, allowing for the detection of molecules at extremely low concentrations through their unique vibrational signatures [48].

#### 3.3.2. Confocal Laser Endomicroscopy

Confocal laser endomicroscopy (CLE) is an advanced imaging technique that enables high-resolution, real-time visualization of histopathology intraoperatively [49]. CLE uses a low-power laser to illuminate tissue, capturing fluorescent light reflected back through a pinhole [50,51]. This setup ensures that both the illumination and detection systems align in the same focal plane, enhancing spatial resolution by excluding out-of-focus light [52]. In application, the process involves directing the laser onto a specific layer of brain tissue; the light that bounces back is recollected by the same lens that directed it. Only light that returns directly through the pinhole is recorded, thereby sharpening the detail and clarity of the resultant image. This allows neurosurgeons to observe cellular details and tissue architecture with precision [49,53].

For optimal imaging, CLE typically requires the administration of intravenous fluorescent agents [49,54]. These markers illuminate the intricate structures within the parenchyma, aiding surgeons in delineating respective boundaries and assessing the health of the tissue in situ [54]. Such detailed visualization supports precise surgical decision-making, crucial for improving surgical outcomes and enhancing patient safety.

#### 3.3.3. Coherent Anti-Stokes Raman Scattering

Coherent anti-stokes Raman scattering (CARS) is a sophisticated imaging technique that leverages the principles of nonlinear optics to provide detailed insights into the molecular composition of materials, including biological tissues [55,56]. It is a type of four-wave mixing process that involves the interaction of two laser beams. When these two intersect in a sample, and the energy difference between them matches the vibrational energy of the material, it induces a vibrational coherence in the molecules of the sample [56,57]. The coherent stimulation of atomic vibrations, where all atoms oscillate in phase, can lead to a significant enhancement of the signal—potentially several orders of magnitude greater than traditional Raman signals, depending on the power of the incident beams and the density of the scatterers [57,58].

CARS has proven highly effective for fast imaging applications, particularly useful for intraoperative imaging [59]. This capability allows for the detailed observation of histological structures in real time without the need for tissue staining [60]. The technique’s ability to provide high-resolution, molecular-selective imaging without photodamage makes it invaluable for investigating biological samples [61]. The penetration depth of a few hundred micrometers offers a significant advantage for surgical guidance, ensuring that functional structures such as blood vessels are not damaged during procedures [36,60]. It is particularly adept at visualizing the distribution and concentration of lipids, which is crucial for studying lipid-rich tissues [59,62].

### 3.4. Applications of Raman Spectroscopy in Neurosurgery

Raman spectroscopy has emerged as a powerful tool for intraoperative guidance during neurosurgical procedures, offering real-time, label-free tissue characterization. Its ability to exploit the unique vibrational signatures of molecules has attracted significant attention in recent years, with numerous studies investigating its potential for tumor margin delineation, tissue classification, and identification of specific tumor types and grades. Table 1 provides a summary of key Raman spectroscopy studies in neurosurgery, highlighting the various device types, key findings, and additional notes for each study. This overview demonstrates the breadth of research in this field and sets the stage for a more detailed discussion of specific applications.

#### 3.4.1. Tumor Margin Delineation

Accurate delineation of tumor margins is crucial for maximizing resection while minimizing damage to healthy brain tissue. Handheld Raman spectroscopy devices have shown promising results in differentiating tumors from normal brain tissue, demonstrating high sensitivity and specificity. Jermyn et al. (2015, 2016) reported that a handheld Raman spectroscopy probe could distinguish normal brain from dense cancer and infiltrated brain with 93% sensitivity and 91% specificity and detect invasive cancer cells up to 3.7 cm beyond the T1-enhanced boundary and 2.4 cm beyond the T2 boundary with 92% accuracy [6,27]. Similarly, Desroches et al. (2015, 2018) characterized a handheld contact Raman spectroscopy probe system and an optical biopsy system using high-wavenumber Raman spectroscopy, achieving 87% accuracy in distinguishing necrotic and vital tissue and 84% accuracy in differentiating dense cancer from non-diagnostic tissue, respectively [2,28].

Advances in data processing and analysis techniques have further improved the performance of Raman spectroscopy for tumor margin delineation. Dallaire et al. (2020) developed a quantitative technique for assessing the quality of Raman spectroscopy measurements, improving tissue classification accuracy from 72% to 80% using only high-quality spectra [14]. Novel applications of Raman spectroscopy, such as the portable visible resonance Raman (VRR-LRRTM) analyzer evaluated by Zhang et al. (2023), have also shown promise, achieving over 80% accuracy in binary classification between normal and tumor tissues [35].

#### 3.4.2. Stimulated Raman Histology (SRH)

Non-handheld systems, such as the NIO Imaging System, have been used for stimulated Raman histology (SRH), providing rapid, label-free, high-resolution images of tissue samples. Studies by Di et al. (2021) demonstrated that SRH provided comparable diagnostic accuracy to frozen sections for intraoperative diagnosis of gliomas (71.4% vs. 76.5%) and meningiomas (100% for both), while significantly reducing the time to diagnosis [17,18]. Einstein et al. (2022) also found no significant difference in diagnostic capability between SRH (78%) and frozen sections (94%) for intraoperative diagnosis of CNS and head and neck tumors [19].

The use of advanced image analysis techniques, such as quantitative SRS microscopy and deep learning, has further enhanced the performance of SRH. Ji et al. (2016) showed that quantitative SRS microscopy could distinguish tumor-infiltrated and non-infiltrated tissue with 97.5% sensitivity and 98.5% specificity [30]. Hollon et al. (2020, 2021) developed an intraoperative diagnostic pipeline combining SRH and deep learning, achieving 94.6% diagnostic accuracy, statistically non-inferior to pathologists’ accuracy using conventional methods, and differentiating glioma recurrence from treatment effects with 95.8% accuracy, 100% sensitivity, and 88.9% specificity [24,25].

#### 3.4.3. Tumor Type and Grade Identification

Raman spectroscopy has shown potential for identifying different tumor types and grades intraoperatively, including both glial and non-glial tumors. For gliomas, Livermore et al. (2019) demonstrated that Raman spectroscopy could classify gliomas into IDH-mutant or IDH-wild-type with 91–95% accuracy and into three genetic subtypes with 79–94% accuracy using fresh tissue [31]. Hollon et al. (2023) developed an AI system combining SRH imaging and deep learning, achieving 93.3% accuracy for predicting IDH mutation, 1p19q co-deletion, and ATRX mutation in diffuse gliomas [26].

The application of Raman spectroscopy extends beyond glial tumors. For meningiomas, Zhang et al. (2022) used a VRR-LRRTM analyzer to identify tumor grades and margins intraoperatively, with SVM classification based on principal component analysis achieving up to 100% accuracy in distinguishing grade I and grade II meningiomas [35]. Di et al. (2021) found that SRH provided 100% diagnostic accuracy for intraoperative diagnosis of meningiomas, comparable to frozen sections [18].

While the majority of studies have focused on gliomas and meningiomas, there is emerging evidence for the utility of Raman spectroscopy in a broader range of tumor types. Einstein et al. (2022) demonstrated the applicability of SRH in diagnosing both CNS and head and neck tumors, finding no significant difference in diagnostic capability between SRH (78%) and frozen sections (94%) [19]. This suggests potential utility for various tumor types, including metastases, although more focused studies on metastatic tumors are needed.

The versatility of Raman spectroscopy in identifying and grading various tumor types highlights its potential as a valuable tool in intraoperative neurosurgery. However, further research is required to validate its effectiveness across a wider range of non-glial tumors, particularly metastases, and to establish standardized protocols for different tumor types.

#### 3.4.4. Integration into Surgical Workflow

The successful integration of Raman spectroscopy into the surgical workflow is crucial for its clinical adoption. Handheld devices offer real-time, in situ tissue characterization, allowing for direct surgical guidance without disrupting the workflow [2,6,23,27]. On the other hand, SRH systems like the NIO require tissue samples to be removed from the patient and placed on glass slides for imaging [14,35,36]. While this process may slightly extend the surgical time, it allows for rapid and accurate tissue assessment by the pathologist. The integration of SRH images into the pathology workflow can facilitate interdisciplinary communication and decision-making [26].

#### 3.4.5. Comparison with Other Techniques

When compared to other intraoperative guidance techniques, Raman spectroscopy has demonstrated competitive performance. Livermore et al. (2021) showed that Raman spectroscopy outperformed 5-ALA fluorescence in differentiating glioma from normal brain tissue, particularly in glioblastoma margin samples, where 5-ALA had a high false-negative rate [32]. However, Herta et al. (2022) found that Raman spectroscopy had higher sensitivity (69% vs. 46%) but lower specificity (57% vs. 81%) compared to 5-ALA for detecting cancer cells in the infiltration zone during glioblastoma resection, suggesting that a combination of the two techniques may offer the best results [22].

#### 3.4.6. Comparison to Current Standard Practice

The studies reviewed in this article demonstrate that intraoperative Raman spectroscopy, particularly stimulated Raman histology (SRH), offers a promising alternative to the current standard practice of frozen histology/pathology. SRH has been shown to provide comparable diagnostic accuracy to frozen sections for intraoperative diagnosis of various brain tumors, including gliomas and meningiomas [14,35,36]. Moreover, SRH significantly reduces the time to diagnosis compared to frozen sections, which can be crucial in the intraoperative setting [14,35].

The combination of SRH with advanced image analysis techniques, such as deep learning, has further enhanced its diagnostic performance. Hollon et al. (2020) developed an intraoperative diagnostic pipeline combining SRH and deep learning, achieving 94.6% diagnostic accuracy, which was statistically non-inferior to pathologists’ accuracy using conventional methods [24]. This suggests that SRH has the potential to streamline the intraoperative diagnostic process, providing rapid and reliable results without compromising accuracy.

Other studies have also shown promising results. Di et al. (2021) found that SRH provided comparable diagnostic accuracy to frozen sections for intraoperative diagnosis of gliomas (71.4% vs. 76.5%) and meningiomas (100% for both) [17,18]. Similarly, Einstein et al. (2022) reported no significant difference in diagnostic capability between SRH (78%) and frozen sections (94%) for intraoperative diagnosis of CNS and head and neck tumors [19].

However, it is important to note that while SRH shows great promise, it is not yet a replacement for conventional histopathology. The studies reviewed here are largely preliminary and have limitations, such as small sample sizes and lack of validation in larger, multi-center trials. Further research is needed to establish the clinical utility and cost-effectiveness of SRH in comparison to the current standard practice, including direct comparisons with pathologist readings across a wider range of tumor types and grades.

#### 3.4.7. Advantages of Raman Spectroscopy in Intraoperative Neurosurgery

Raman spectroscopy offers numerous advantages in the realm of intraoperative neurosurgery, demonstrating distinct benefits in the resection of brain tumors. Firstly, it provides molecular information to the neurosurgeon and operative team in real time, allowing for increased accuracy in tumor resection by differentiating healthy and cancerous tissue more effectively than traditional methods, such as visual inspection and palpation [7]. The precision offered by Raman spectroscopy helps maximize the extent of resection and improve overall outcomes. Secondly, Raman spectroscopy can lead to a reduction in surgery time as it allows neurosurgeons to quickly identify tumor boundaries without the need for extensive tissue sampling and pathological analysis. Furthermore, this reduction in surgery time benefits the patients by minimizing the duration of anesthesia exposure to the patient, decreasing its associated risks [22]. When compared to other intraoperative techniques, such as intraoperative MRI or fluorescence-guided surgery, Raman spectroscopy stands out for its non-destructive nature and high specificity, allowing for precise tumor delineation without altering tissue integrity [10,21]. Lastly, its compatibility with existing surgical approaches and instruments positions it to be a powerful intraoperative aid without disrupting the surgical workflow [21].

## 4. Discussion

### 4.1. Future Directions and Limitations

Despite the promising results demonstrated by intraoperative Raman spectroscopy in neurosurgical applications, several limitations and future directions should be considered. Many of the studies have small sample sizes and lack validation in larger, multi-center trials. The long-term impact of Raman spectroscopy-guided surgery on patient outcomes and survival requires further investigation. Technological advancements are necessary to streamline the integration of Raman spectroscopy into the surgical workflow, including the development of miniaturized, robust, and cost-effective devices. The combination of Raman spectroscopy with other imaging modalities, such as fluorescence and optical coherence tomography, may provide a more comprehensive understanding of tissue composition and structure.

Future studies in Raman spectroscopy for intraoperative neurosurgery are necessary for both handheld Raman optical biopsy probe technology and non-handheld stimulated Raman histology. Regarding handheld technology, the Raman spectroscopy optical probe technology generally requires further verification and additional clinical trials concerning intraoperative use. Additional studies are also required to understand the potential of Raman spectroscopy optical probes for glioma subtype differentiation and to evaluate the effect of intraoperative blood on Raman spectroscopy grade classification of gliomas [16]. Regarding stimulated Raman histology, future areas of focus include improved tumor classification and integration of stimulated Raman histology with molecular and clinical findings. While stimulated Raman histology has been utilized to identify meningiomas, further studies are required for meningioma-grade classification [35]. Additionally, while stimulated Raman histology has been utilized for molecular classification of gliomas, additional data collection, model training, and validation are needed for a robust combined histopathological and molecular classification of gliomas [31].

### 4.2. Ethical Considerations and Patient Safety

The introduction of a new technology in surgical practice should consider a variety of ethical considerations regarding the feasibility and necessity of the technology, regulatory and clinical approval, and patient safety. The current literature indicates that the use of Raman spectroscopy in neurosurgery, specifically in CNS tumor management, provides utility in diagnosis, tumor margin delineation, and surgical treatment. Such studies highlight a need in the field that Raman spectroscopy fulfills. However, further studies should still be conducted to investigate the extent of its potential through in vivo studies as most studies to date are primarily ex vivo [29]. Validation studies will not only allow for a more robust comparison with current standards of treatment and diagnoses but also ensure safety. Building off the importance of safety and efficiency, as many studies still lack data on the long-term effects and have large rates of loss to follow-up, a systematic method for risk–benefit analysis should also be considered to determine patient candidacy for the technology. Due to Raman spectroscopy’s non-destructive and high-precision nature, the risks should be low but still considered. Through consideration of all such aspects, cost–benefit analysis and integration into clinical practice after regulatory approval will be better streamlined.

## 5. Conclusions

In conclusion, intraoperative Raman spectroscopy has shown great potential for tumor margin delineation, tissue classification, and identification of specific tumor types and grades in neurosurgery. Handheld devices and SRH systems offer unique advantages and can be integrated into the surgical workflow to guide decision-making. While SRH demonstrates comparable accuracy to frozen sections and significantly reduces the time to diagnosis, further research is needed to establish its clinical utility and cost-effectiveness compared to the current standard practice. Larger clinical studies and continued technological advancements are necessary to fully realize the potential of Raman spectroscopy in the neurosurgical operating room.

## Figures and Tables

**Figure 1 biomedicines-12-02363-f001:**
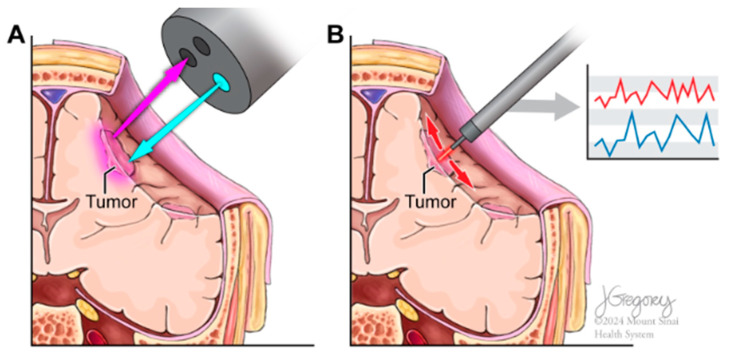
(**A**). An intraoperative Raman spectroscopy probe emits a laser onto brain tissue during surgery, differentiating between tumor and healthy tissue in real-time. The laser light interacts with molecular structures, generating a Raman signal that reveals biochemical differences. (**B**). The Raman spectrum displays distinct frequency peaks that differentiate tumor tissue from the surrounding normal brain parenchyma. The clear separation of spectral signatures highlights the effectiveness of Raman spectroscopy in identifying pathological tissues during surgery.

**Figure 2 biomedicines-12-02363-f002:**
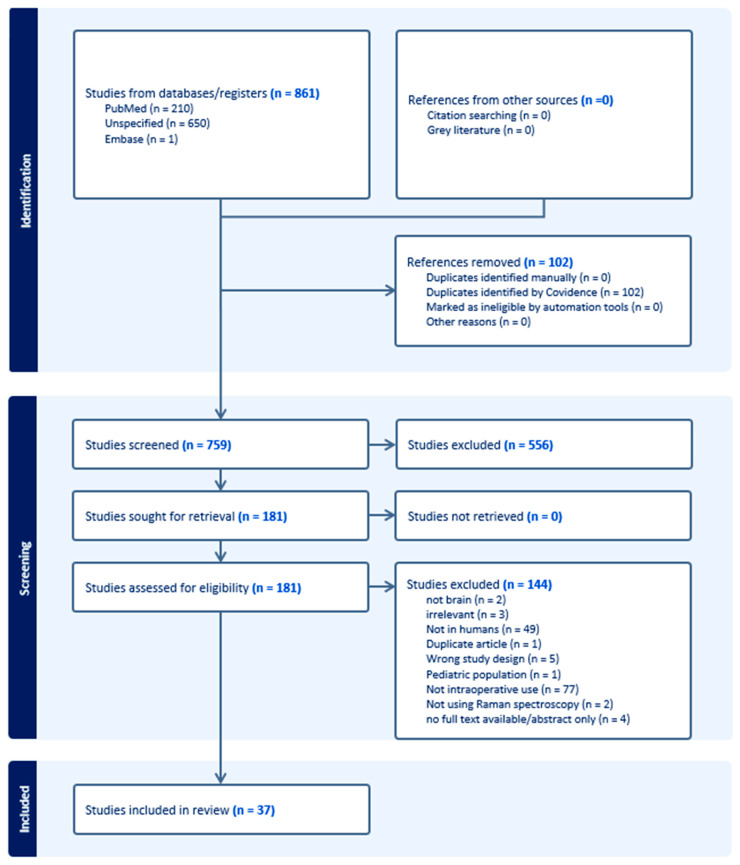
PRISMA summary and schematic representation of the systematic literature review of manuscripts pertinent to neuronavigation.

**Figure 3 biomedicines-12-02363-f003:**
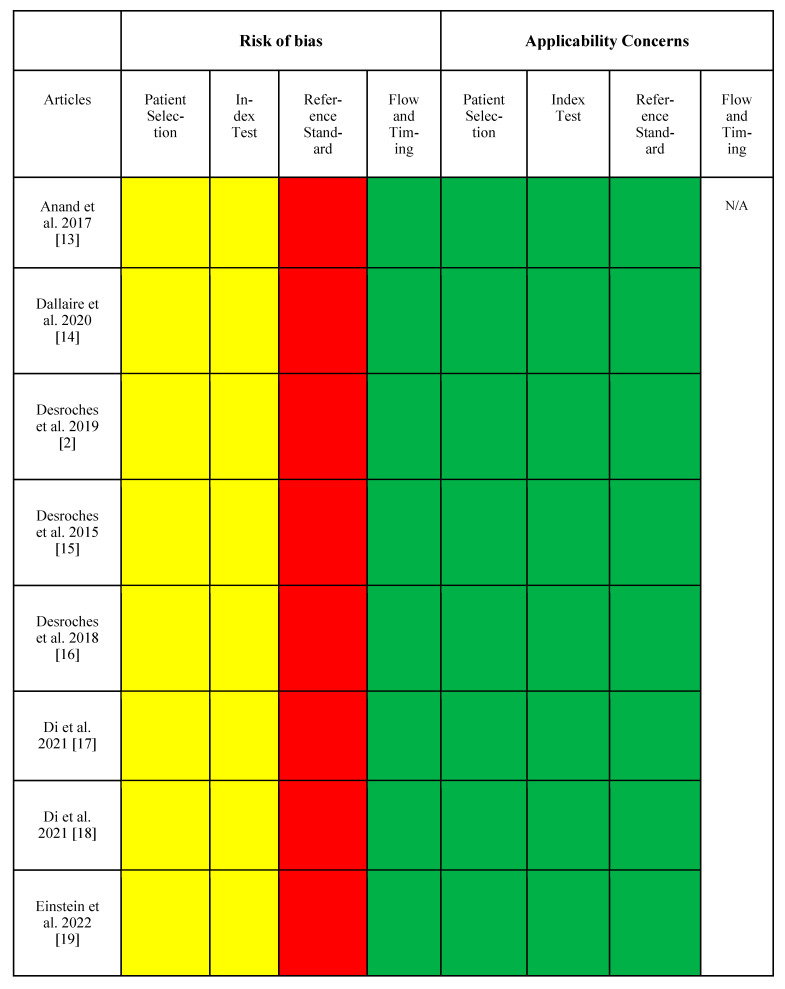
QUADAS-2 risk of bias and applicability assessment for studies evaluating the diagnostic accuracy of Raman spectroscopy and multimodal fiber–probe techniques. The references cited in the figure includes [2,6,10,13,14,15,16,17,18,19,20,21,22,23,24,25,26,27,28,29,30,31,32,33,34,35,36].

**Table 1 biomedicines-12-02363-t001:** Summary of key studies on Raman spectroscopy applications in intraoperative neurosurgery (2015–2023). This table presents a comprehensive overview of significant research studies investigating the use of Raman spectroscopy techniques in neurosurgical procedures. It highlights various device types, including handheld probes and non-handheld systems, along with their performance in tumor detection, margin delineation, and classification. The studies demonstrate the evolving capabilities of Raman spectroscopy, from basic tissue differentiation to advanced molecular classification, and its comparison with current standard practices. This summary illustrates the potential of Raman spectroscopy to enhance surgical precision and improve patient outcomes in neurosurgery.

Author/Year	Device Type	Key Findings	Additional Notes
Jermyn et al., 2015, 2016, 2017 [6,27,28]	Handheld Raman probe	- 93% sensitivity and 91% specificity in distinguishing normal brain from dense cancer and infiltrated brain - Detected invasive cancer cells up to 3.7 cm beyond the T1-enhanced boundary and 2.4 cm beyond the T2 boundary with 92% accuracy - Highly accurate detection of cancer in situ with multimodal optical spectroscopy	Demonstrated potential for improving tumor margin delineation in various glioma grades and metastatic cancers
Desroches et al., 2015, 2018, 2019 [2,15,16]	Handheld contact Raman probe	- 87% accuracy in distinguishing necrotic and vital tissue - 84% accuracy in differentiating dense cancer from non-diagnostic tissue - First in-human use of a Raman spectroscopy guidance system integrated with a brain biopsy needle	Characterized handheld systems and an optical biopsy system using high-wavenumber Raman spectroscopy
Dallaire et al., 2020 [14]	Not specified	Improved tissue classification accuracy from 72% to 80% using only high-quality spectra	Developed a quantitative technique for assessing the quality of Raman spectroscopy measurements
Zhang et al., 2023 [35]	Portable visible resonance Raman (VRR-LRRTM) analyzer	Over 80% accuracy in binary classification between normal and tumor tissues	Novel application of visible resonance Raman spectroscopy for gliomas
Zhang et al., 2022 [36]	VRR-LRRTM analyzer	Up to 100% accuracy in distinguishing grade I and grade II meningiomas	Focused on intraoperative detection of meningiomas
Di et al., 2021 [17,18]	NIO Imaging System (SRH)	- Comparable diagnostic accuracy to frozen sections for gliomas (71.4% vs. 76.5%) and meningiomas (100% for both) - Significantly reduced time to diagnosis	Demonstrated the potential of SRH as an alternative to frozen sections for both gliomas and meningiomas
Einstein et al., 2022 [19]	NIO Imaging System (SRH)	No significant difference in diagnostic capability between SRH (78%) and frozen sections (94%) for CNS and head and neck tumors	Further validated SRH’s performance compared to standard practice
Ji et al., 2015 [30]	Quantitative SRS microscopy	97.5% sensitivity and 98.5% specificity in distinguishing tumor-infiltrated and non-infiltrated tissue	Showed the potential of advanced SRS techniques in 22 neurosurgical patients
Hollon et al., 2018, 2020, 2021, 2023 [23,24,25,26]	Stimulated Raman Histology (SRH) with deep learning	- 94.6% diagnostic accuracy, statistically non-inferior to pathologists’ accuracy using conventional methods - 95.8% accuracy in differentiating glioma recurrence from treatment effects - 93.3% accuracy for predicting IDH mutation, 1p19q co-deletion, and ATRX mutation in diffuse gliomas	Combined SRH with deep learning for improved performance in various applications, including pediatric brain tumors and molecular classification
Livermore et al., 2019 [31]	Raman spectroscopy (type not specified)	- 91–95% accuracy in classifying gliomas into IDH-mutant or IDH-wild-type - 79–94% accuracy in classifying gliomas into three genetic subtypes	Demonstrated potential for molecular classification of gliomas
Livermore et al., 2021 [32]	Raman spectroscopy (type not specified)	Outperformed 5-ALA fluorescence in differentiating glioma from normal brain tissue, particularly in glioblastoma margin samples	Comparative study with 5-ALA fluorescence
Herta et al., 2023 [22]	Raman spectroscopy (type not specified)	Higher sensitivity (69% vs. 46%) but lower specificity (57% vs. 81%) compared to 5-ALA for detecting cancer cells in the glioblastoma infiltration zone	Suggested potential benefits of combining Raman spectroscopy with 5-ALA
Galli et al., 2019 [21]	Near-infrared Raman and fluorescence spectroscopy	Able to distinguish neoplastic from non-neoplastic tissue and differentiate glioma from metastases	A study of 209 patients; showed potential for diagnostics but noted limitations in tumor type recognition due to necrosis or normal tissue presence
Jabarkheel et al., 2022 [10]	Raman spectroscopy with machine learning	AUC of 0.94 for tumor vs. normal brain and 0.91 for LGG vs. normal brain	Focused on rapid intraoperative diagnosis of pediatric brain tumors
Jiang et al., 2022 [29]	NIO Imaging System (SRH) with AI	96.6% overall diagnostic accuracy for skull base tumors using supervised contrastive learning	Demonstrated ability to segment tumor-normal margins and detect microscopic tumor infiltration
Anand et al., 2017 [13]	Multimodal fiber–probe spectroscopy	Effective in detecting epileptogenic focal cortical dysplasia in children	Combined fluorescence and Raman spectroscopy
Fitzgerald et al., 2022 [20]	NIO Imaging System (SRH)	93.3% overall accuracy for sinonasal and skull base tumors, with significantly faster diagnosis time compared to frozen sections	Demonstrated SRH’s utility in a variety of skull base tumor types

## Data Availability

All data utilized in this study exist.

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
