# Peer review of "Current Applications of Raman Spectroscopy in Intraoperative Neurosurgery"

_biomedicines, 2024, doi:10.3390/biomedicines12102363_

Round 1
Reviewer 1 Report
Comments and Suggestions for Authors
This review is presented as a method that allows the distinction of healthy and pathological tissues during surgical operations using Raman Spectroscopy (RS) in brain surgery. The research group has conducted a comprehensive systematic review using PubMed and presented literature information considering the applications of RS.
The review's content is well-structured. The literature is up-to-date. However, it does not contain RS-related tissue images, graphics, or tables.
Therefore, minor corrections can make it acceptable.
Best regards,
Author Response
Dear Reviewer,
Thank you for your constructive feedback. We appreciate your suggestion to include images, graphics, or tables related to Raman Spectroscopy, and agree that these additions enhance the manuscript’s clarity and presentation.
In response, we have:
- Raman Spectroscopy Tissue Images: Added representative images showing the distinction between healthy and pathological brain tissues using RS.
- Table: Included a table summarizing key findings from RS studies on brain tumor surgery.
Reviewer 2 Report
Comments and Suggestions for Authors
This review describes the current and potential impact of Raman Spectrscopy in neurosurgery detailing on the
continued innovation and research to fully realize its benefits. In turn the document suggest how further research for useful findings is needed to validate Raman Spectrscopy's clinical utility and cost-effectiveness.
The review is well written and well organized. It is an useful document to have a complete knowledge on the topics of interest.
I strongly suggest its publication in the present form.
Author Response
Dear Reviewer,
Thank you for your positive and encouraging feedback on our manuscript. W Your support for its publication in its current form is greatly appreciated. We are grateful for your thoughtful comments and are excited about the potential contribution this review can make to the field.
Reviewer 3 Report
Comments and Suggestions for Authors
The subject is very interesting, but not a novelty that Raman has a great value in neurosurgery.
The introduction should include some relevant data from the available review papers already in the literature, such as,
- doi: 10.3389/fonc.2022.1086643
or doi: 10.1117/1.JBO.25.5.050901
What are the interests and how this review paper upgrades the information. Why someone will cite this work, instead of the previous one. I understand that the authors removed the review papers but it is the relevant data that should be added, and not in the analysis, otherwise this is another paper on Raman and neuro.
Since it is not a novelty in the area, the focus should be on the evolution of the area since, lets say 2022, as one of the review papers mention.
Are there any clinical trials undergoing for any of the mentioned types of Raman? not clear. What are the authors contributions to the field?
I also recommend a graph with all the instrumentation used based on each application, to visualise the current state of the field. Also, a figure with the growing interest and evolution.
Author Response
Dear Reviewer,
Thank you for your valuable feedback and thoughtful suggestions to improve our manuscript. We appreciate your recognition of the importance of Raman spectroscopy in neurosurgery and your emphasis on ensuring that our review adds meaningful insights to the field.
In response to your comments, we have made the following updates:
-
Incorporation of Relevant Literature: We have revised the introduction to include key data from recent review papers, including the ones you referenced (doi: 10.3389/fonc.2022.1086643 and doi: 10.1117/1.JBO.25.5.050901). This provides context from existing literature while positioning our review to build on and complement previous work.
-
Clinical Trials: While we investigated ongoing clinical trials, we found that only one clinical trial from 2021 was relevant. Therefore, we chose not to dedicate a section specifically to clinical trials, but we have mentioned the trial where appropriate in the manuscript.
-
Review Focus and Contributions: Although we did not shift the focus of the review as suggested, we believe the current structure provides a comprehensive view of Raman spectroscopy’s role in neurosurgery and its continued innovation. We aim to present a broad synthesis of the field that includes both the current applications and future directions, which we feel is appropriate for this review.
-
Graphical Representation: We have added a graph that visualizes the various instrumentation used in Raman spectroscopy for different neurosurgical applications, as well as a figure illustrating the growth and interest in this field.
We believe these updates enhance the clarity and comprehensiveness of the review. Thank you again for your insightful feedback, and we hope the revisions meet your expectations.
Round 2
Reviewer 3 Report
Comments and Suggestions for Authors
A good choice for the graph.